# Object Representations as Fixed Points: Training Iterative Refinement Algorithms with Implicit Differentiation

**Michael Chang**[*], **Thomas L. Griffiths**[†], **Sergey Levine**[*]
[*]University of California, Berkeley, [†]Princeton University
mbchang@berkeley.edu, tomg@princeton.edu, svlevine@eecs.berkeley.edu

## Abstract

Iterative refinement – start with a random guess, then iteratively improve the guess – is a useful paradigm for representation learning because it offers a way to break symmetries among equally plausible explanations for the data. This property enables the application of such methods to infer representations of sets of entities, such as objects in physical scenes, structurally resembling clustering algorithms in latent space. However, most prior works differentiate through the unrolled refinement process, which can make optimization challenging. We observe that such methods can be made differentiable by means of the implicit function theorem, and develop an implicit differentiation approach that improves the stability and tractability of training by decoupling the forward and backward passes. This connection enables us to apply advances in optimizing implicit layers to not only improve the optimization of the slot attention module in SLATE, a state-of-the-art method for learning entity representations, but do so with constant space and time complexity in backpropagation and only one additional line of code. [1]

## 1 Introduction

Conventionally, neural network models implement feedforward computation, transforming the input $x$ into the output $z$ through a fixed series of operations corresponding to distinct layers as $z = f_N(f_{N-1}(...f_2(f_1(x))...))$. However, a range of more sophisticated models implement *iterative* computation with the network, typically formulated and motivated as some sort of optimization procedure where the correct answer is the fixed point of an iterative refinement $z = f_x(z)$ of an initial guess $z_0$. This includes diffusion models [30, 52, 58, 59], energy-based models [14, 37], deep equilibrium models [7], iterative amortized inference procedures [40–42], neural ordinary differential equations [11], meta-learning algorithms [5, 17, 23], and object-centric models [16, 35, 61, 62]. Such iterative refinement procedures may have a number of advantages over direct feedforward computation: they can serve to simplify the learned function (e.g., in the same way that a recursive program might be much simpler than an equivalent program implemented without recursion), introduce an inductive bias into the model that improves generalization, and break symmetries.

In this work, we consider the case of iterative refinement applied to representation learning of latent sets, which has primarily been applied to learning representations of objects from pixels [24–26, 35, 39, 57, 61, 62, 70]. The particular challenge of this setting is that invariance of set elements to permutation means there are many latent sets that serve as equally plausible explanations for the data. Iterative refinement is especially useful in this context because it breaks the symmetry among these explanations with the randomness of the initial guess $z_0$ rather than encoding the symmetry-breaking mechanism in the weights of the network, as do conventional methods that learn a direct feedforward

---

[1]Project website: https://sites.google.com/view/implicit-slot-attention.

36th Conference on Neural Information Processing Systems (NeurIPS 2022).

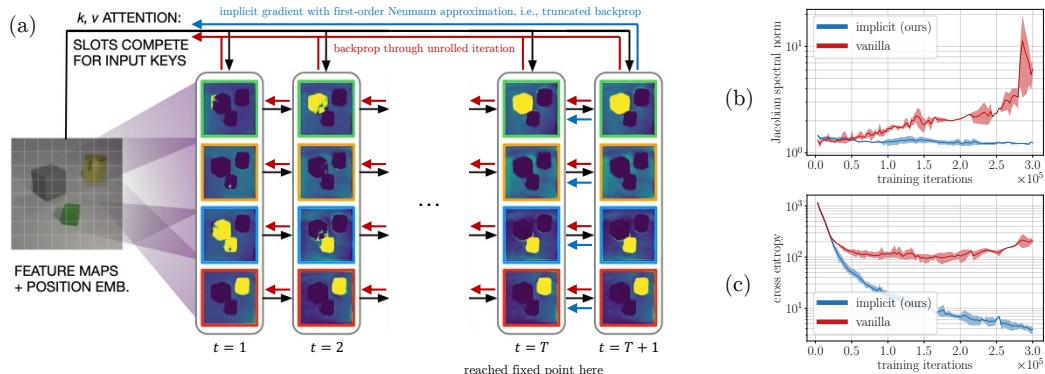

Figure 1: **Overview.** We address efficient training of iterative refinement methods for learning representations of latent sets, such as the slot attention model in (a), whose illustration is adapted from Locatello et al. [39]. Vanilla slot attention backpropagates gradients through the unrolled iterative refinement procedure, which leads to training instabilities as shown by the growing Jacobian spectral norm (b). Implicit slot attention uses the first-order Neumann approximation of the implicit gradient, which simply truncates the backpropagation, leading to substantially more effective training, as shown in (c).

mapping from observation to representation. The state-of-the-art of these iterative object-centric methods is the slot attention module from Locatello et al. [39], serving as the focus of our paper.

Unfortunately, these methods have been notoriously difficult to train. Their nature as unsupervised representation learning methods means that we do not have ground-truth targets to supervise the outputs of each iteration of $f$, as in other settings (e.g. diffusion models). As a result, prior works train these methods by differentiating through the unrolled iterations of $f$. Differentiating through this recurrence contributes to various training instabilities, as we can see by the growing spectral norm of $f$ in Fig. 1b. Such instabilities result in sensitivity to hyperparameter choices (e.g., number of refinement steps) and have motivated adding optimization tricks such as gradient clipping, learning rate warm-up, and learning rate decay, all of which make such models more complex and harder to use, restrict the model from optimizing its learning objective fully, and only temporarily delay instabilities that still emerge in later stages of training.

To approach this problem, we observe that previous iterative refinement methods like slot attention have not taken full advantage of the fact that $f$ can be viewed as a fixed point operation. Thus, $f$ can be trained with **implicit differentiation** applied at the fixed point, without backpropagating gradients through the unrolled iterations [9, 15]. This paper investigates how advances in implicit differentiation in neural models can be applied to improve the training of iterative refinement methods.

Our primary contribution is to propose implicit differentiation for training iterative refinement procedures, specifically slot attention, for learning representations of latent sets. First, we show that slot attention can be cast as a fixed point procedure that can be trained with implicit differentiation, resulting in what we call **implicit slot attention**. Second, we show on the latest state-of-the-art method of this kind, SLATE [57], that using the first-order Neumann approximation of the implicit gradient for the slot attention module yields substantial improvement in optimization. Third, we show across three datasets that, compared to SLATE, our method for training achieves much lower validation loss in training, as well as lower Fréchet inception distance (FID) [29] and mean squared error (MSE) in image reconstruction. Fourth, our method also removes the need for gradient clipping, learning rate warmup, or tuning the number of iterations, while achieving lower space and time complexity in the backward pass, all with just one additional line of code. Fifth, when integrated with the original slot attention encoders and decoders from Locatello et al. [39], implicit differentiation substantially improves object property prediction and continues to predict intuitive segmentation masks as the vanilla slot attention.

## 2   Related Work

Much early work in artificial intelligence followed a paradigm of using an iterative learning procedure during execution time, whether it be a form of search, inference, or optimization [51]. These include

early work on variational inference [12, 19] and energy-based models [14, 37, 65–68], Hopfield networks [31], Boltzmann machines [1], and associative memory [36]. The rise of modern deep networks in the last decade shifted the method for computing solutions during execution time away from iterative procedures and towards producing the solution directly with a feedforward pass of a network. Recent works have started to shift the paradigm of execution back to combining the best of function approximation and iterative search, with neural networks parameterizing initializations [17], update rules [5, 24, 40], search heurstics [34, 56], and evaluation functions [14]. Our work concerns the optimization of neural networks as update rules for these iterative refinement procedures.

Our methodological novelty is the adaptation of implicit differentiation techniques for training iterative refinement procedures for representing of latent sets, where our key insight is that the iterative procedure used for symmetry-breaking reaches a fixed point, thus allowing implicit differentiation to be used. We are not aware that this has been done before, as current methods that perform iterative refinement for representing latent sets, often referred to as **object-centric learning** [24–26, 35, 39, 57, 61, 62, 70], all differentiate through the unrolled dynamics of the fixed point procedure, which as we show makes them difficult to train.

Our work draws upon innovations in implicit differentiation that have been applied in various other applications besides object-centric learning, such as embedded optimization layers [2, 4], neural ordinary differential equations [11], meta-learning [49], implicit neural representations [32], declarative layers [22], and transformers [7]. Although we evaluate various techniques for implicit differentiation, our results highlight the benefits of the simplest of these, which is that of using a truncated Neumann approximation [20, 21, 32, 55]. While this technique was used in Zoran et al. [70] without theoretical explanation, we propose implicit differentiation as an explanation for why this technique is beneficial. The closest works to ours is the concurrent work of Zhang et al. [69] which applies implicit differentiation to predicting properties of set elements and finds similar benefits. Our approach generalizes their results to the unsupervised setting and scales to high dimensional outputs (e.g., images), taking a crucial step toward improving representation learning of latent sets.

## 3  Background

Our work builds on prior works on iterative refinement and implicit differentiation with deep networks.

**Iterative refinement for inferring representations of latent sets**  Current work on inferring representations of latent sets is motivated by learning to represent objects – which we consider *independent* and *symmetric* entities – from perceptual input. Because mixture models are also defined to have *a priori* independent and symmetric mixture components, they have been the model of choice for representing entities: thus these iterative methods model each datapoint $x^n$ (e.g. image) as a set of independent sensor measurements $x^{n,m}$ (e.g. pixels) which are posited as having been generated from a mixture model whose components represent the entities. Under a clustering lens, the problem reduces to finding the $K$ groups of cluster parameters $\boldsymbol{\theta}^n := \{\theta^{n,k}\}_{k=1}^K$ and cluster assignments $\boldsymbol{\phi}^{n,m} := \{\phi^{n,m,k}\}_{k=1}^K$ that were responsible for the measurements $x^{n,m}$ of the datapoint $x^n$.

A network $f$ breaks symmetry among components by alternately updating $\boldsymbol{\theta}^n$ and $\boldsymbol{\phi}^{n,m}$ starting from independent randomly initialized $\theta^{n,k}$'s. The slot attention module [39], e.g., computes $\boldsymbol{\theta}_{t+1}^n \leftarrow f(\boldsymbol{\theta}_t^n, x^n)$, where $\boldsymbol{\phi}^{n,m}$ is updated as an intermediate step inside $f$. The $\boldsymbol{\theta}^n$, called *slots*, serve as input to a downstream objective, e.g. image reconstruction, whose gradients are backpropagated through the unrolling of $f$. Earlier works applied this approach to binary images [24] and videos [61].

**Implicit differentiation**  Implicit differentiation is a technique for computing the gradients of a function defined in terms of satisfying a joint condition on the input and output. For example, a fixed point operation $f$ is defined to satisfy "find $z$ such that $z = f(z, x)$" (or written as $z = f_x(z)$) rather than through an explicit parameterization of $f$. This fixed point $z_*$ can be computed by simply repeatedly applying $f$ or by using a black-box root-finding solver. Letting $f_\mathbf{w}$ be parameterized by weights $\mathbf{w}$, with input $x$ and fixed point $z_*$, the implicit function theorem [9] enables us to directly compute the gradient of the loss $\ell$ with respect to $\mathbf{w}$, using only the output $z_*$:

$$\frac{\partial \ell}{\partial \mathbf{w}} = \underbrace{\frac{\partial \ell}{\partial z_*} (I - J_{f_\mathbf{w}}(z_*))^{-1}}_{\mathbf{u}^\top} \frac{\partial f_\mathbf{w}(z_*, x)}{\partial \mathbf{w}}, \tag{1}$$

where $J_{f_{\mathbf{w}}}(z_*)$ is the Jacobian matrix of $f_{\mathbf{w}}$ evaluated at $z_*$. Compared to backpropagating through the unrolled iteration of $f$, which is just one of many choices of the solver, implicit differentiation via Eq. 1 removes the cost of storing any intermediate results from the unrolled iteration. Deep equilibrium models (DEQ) [7] represent the class of functions $f$ parameterized by neural networks, which have successfully been trained with implicit differentiation and empirically produce stable fixed points even though their convergence properties remains theoretically not well understood.

Much effort has been put into approximating the inverse-Jacobian term $(I - J_{f_{\mathbf{w}}}(z_*))^{-1}$ which has $\mathcal{O}(n^3)$ complexity to compute. Using notation from Bai et al. [8], Pineda [48] and Almeida [3] propose to approximate the $\mathbf{u}^{\top}$ term in Eq. 1 as the fixed point of the linear system:

$$\mathbf{u}^{\top} = \mathbf{u}^{\top} J_{f_{\mathbf{w}}}(z_*) + \frac{\partial \ell}{\partial z_*}, \tag{2}$$

which can be solved with any black-box solver. However, in the context of applying implicit differentiation to neural networks, this approach has in practice required some expensive regularization to maintain stability [8], so Fung et al. [20], Geng et al. [21], Huang et al. [32], Shaban et al. [55] propose instead to approximate $(I - J_{f_{\mathbf{w}}}(z_*))^{-1}$ with its Neumann series expansion, yielding

$$\frac{\partial \ell}{\partial \mathbf{w}} = \lim_{T \to \infty} \frac{\partial \ell}{\partial z_*} \sum_{i=0}^{T} J_{f_{\mathbf{w}}}(z_*)^i \frac{\partial f_{\mathbf{w}}(z_*, x)}{\partial \mathbf{w}}. \tag{3}$$

The first-order approximation ($T = 1$) amounts to applying $f$ once to the fixed point $z_*$ and differentiating through the resulting computation graph. This is not only cheap to compute and easy to implement, but has also been shown empirically [21] to have a regularizing effect on the spectral norm of $J_{f_{\mathbf{w}}}$ without sacrificing performance.

## 4 Implicit Iterative Refinement

Our main contribution is to propose treating iterative refinement algorithms used for inferring latent sets as fixed-point procedures, thereby motivating the use of implicit differentiation to train them.

### 4.1 Motivation

Our approach is inspired by the structural resemblance between iterative refinement for latent sets and the Expectation-Maximization algorithm [13], which also has been pointed out by Greff et al. [24] and Locatello et al. [39]. If $f$ were to update the components $\boldsymbol{\theta}^n$ and assignments $\phi^{n,m}$ in a way that monotonically improves the evidence lower bound (ELBO) $\mathcal{L}^n$ on the log-likelihood of each image $x^n$ with respect to $\boldsymbol{\theta}^n$ and $\phi^{n,m}$, then we know from Neal and Hinton [43] and Wu [64] that such an approach is a fixed point operation whose fixed point locally maximizes $\mathcal{L}^n$. With this interpretation, such iterative refinement algorithms for inferring latent sets can be viewed as performing a nested optimization with three levels, where the weights of the slot attention module are optimized across images $x^n$, the components $\boldsymbol{\theta}^n$ are optimized per-image $x^n$ but across measurements $x^{n,m}$, and the assignments $\phi^{n,m}$ are optimized per-measurement $x^{n,m}$.

How these iterative refinement methods are implemented differs from optimizing the per-datapoint ELBO described above, however. Indeed, both Greff et al. [24] and Locatello et al. [39] implemented ablations to their methods that do implement a variant of Expectation-Maximization on a well-defined mixture model, but found that replacing the update rule with a recurrent neural network empirically performs better at extracting representations.

### 4.2 Iterative refinement as a fixed point procedure

While in general we still lack the theory to understand whether and how the slots of slot attention optimize a well-defined objective like the ELBO, and thus have no theoretical guarantees that slot attention does converge to a fixed point, we empirically observe from its stable forward relative residuals (Fig. 3a) that it does appear to approximately converge to a fixed point. This suggests that slot attention can be understood as an instance of a DEQ that uses naive forward iteration to find a fixed point and backpropagates through the iteration to compute the gradient. The novel implication of this to inferring latent sets is that *any* root-finding solver and implicit gradient estimator can in

```
def step(slots, k, v):                              def iterate(f, x, num_iters):
    # compute assignments given slots                   for _ in range(num_iters):
    q = project_q(norm_slots(slots))                        x = f(x)
    k = k * (slot_size ** (-0.5))                       return x
    attn = F.softmax(torch.einsum('bkd,bqd->bkq', k, q), dim=-1)
    attn = attn / torch.sum(attn + epsilon, dim=-2, keepdim=True)    def forward(inputs, slots):
    # update slots given assignments                    inputs = norm_inputs(inputs)
    updates = torch.einsum('bvq,bvd->bqd', attn, v)     k, v = project_k(inputs), project_v(inputs)
    slots = gru(updates, slots)                         slots = iterate(lambda z: step(z, k, v), slots, num_iterations)
    slots = slots + mlp(norm_mlp(slots))                slots = step(slots.detach(), k, v)
    return slots                                        return slots
```

Figure 2: **Code.** The first order Neumann approximation to the implicit gradient adds only one additional line of Pytorch code [45] to the original forward function of slot attention, but yields substantial improvement of optimization. `attn` and `slots` correspond to $\phi$ and $\theta$ in the text respectively.

principle be used to train slot attention, and we find that many combinations do train slot attention effectively in Fig. 3b. This insight is one of our main contributions, but in the next section we describe a particular combination that we empirically found effective.

### 4.3 Implicit slot attention

We propose **implicit slot attention**: a method for training the state-of-the-art slot attention module [39] with the simplest and most effective method that we have empirically found for approximating the implicit gradient, which is its first-order Neumann approximation (Eq. 3). It can be implemented by simply differentiating the computation graph of applying the slot attention update *once* to the fixed point $\theta_*^n$, where $\theta_*^n$ is computed by simply iterating the slot attention module forward as usual, but *without* the gradient

Table 1: Complexity

|                   | vanilla | implicit |
|-------------------|---------|----------|
| time (forward)    | $\mathcal{O}(n)$ | $\mathcal{O}(n)$ |
| space (forward)   | $\mathcal{O}(n)$ | $\mathcal{O}(1)$ |
| time (backward)   | $\mathcal{O}(n)$ | $\mathcal{O}(1)$ |
| space (backward)  | $\mathcal{O}(n)$ | $\mathcal{O}(1)$. |

tape, amounting to truncating the backpropagation. The time and space complexity of backpropagation for implicit slot attention compared to vanilla slot attention as a function of the number of slot attention iterations $n$, is shown in Table 1. While other methods for implicit differentiation are more complex to implement, this method requires only one additional line of code (Fig. 2).

## 5 Experiments

Our main hypothesis is that iterative refinement methods for latent sets, specifically slot attention, can be trained as DEQs, and that consequently implicit differentiation would improve their optimization. We test this premise by replacing the backward pass of the slot attention module in the state-of-the-art SLATE [57] with the first-order Neumann approximation of the implicit gradient. This requires only one additional line of code to the original slot attention implementation (Fig. 2) and in one instance improves reconstruction mean-squared error by almost 7x.

### 5.1 Experimental setup

We summarize the SLATE architecture, datasets we considered, and relevant implementation details.

**SLATE** SLATE uses a discrete VAE [50] to compress an input image into a grid of discrete tokens. These tokens index into a codebook of latent code-vectors, which, after applying a learned position encoding, serve as the input to the slot attention module. An Image GPT decoder [10] is trained with a cross-entropy loss to autoregressively reconstruct the latent code-vectors, using the outputted slots from slot attention as queries and the latent code-vectors as keys/values. Gradients are blocked from flowing between the discrete VAE and the rest of the network (i.e. the slot attention module and the Image GPT decoder), but the entire system is trained simultaneously.

**Data** We consider three datasets: CLEVR-Mirror [57], Shapestacks [27], and COCO-2017 [38], the former two of which were used in the original SLATE paper. We obtained CLEVR-Mirror directly from the SLATE authors and used a 70-15-15 split for training, validation, and testing. We pooled all the data variants of Shapestacks together as Singh et al. [57] did and used the original train-validation-test splits. The COCO-2017 dataset was downloaded from FiftyOne and used the original train-validation-test splits.

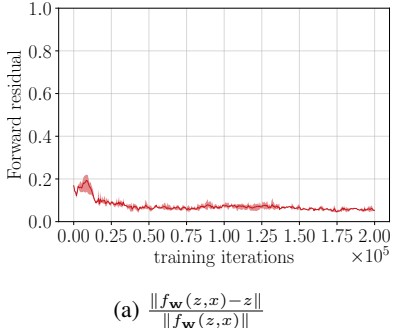
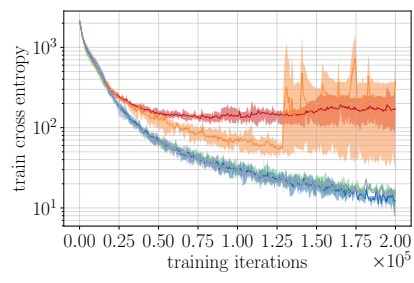

(a) $\frac{\|f_{\mathbf{w}}(z,x)-z\|}{\|f_{\mathbf{w}}(z,x)\|}$    (b) Different solvers

Figure 3: **Can slot attention by trained as a fixed point operation with implicit differentiation?** (3a) The relative residual of the forward iteration of slot attention is close to zero, which motivates its treatment as a fixed point operation. (3b) The forward and backward computations of vanilla slot attention can be swapped out with different solvers, most of which result in the same improved optimization performance. Here we compare with *IB*, *BB*, *IN*, and *BN* variants of implicit differentiation across 4 seeds. Table 2 defines these acronyms.

**Implementation details** In all of our experiments we used the same model and training hyper-parameters as those used for the ShapeStacks experiment from Singh et al. [57, Table 6], run on an A100 GPU. The only difference is in the image resolution (and consequently number of image tokens) because the largest image size we could train with was 96x96 due to computing constraints, rather than the 128x128 used for their CLEVR experiment. However, as we show for resolutions of both 96x96 and 64x64, the improvement in optimization appears to be hold across image resolutions.

For a clean comparison, we simply integrated the official SLATE implementation[2] with the solvers and backward gradient hook from the official implementation of deep equilibrium models[3]. Besides implementing logging and evaluation code, as well as the first-order Neumann approximation (Eq. 3), we did not change anything in the above two official implementations otherwise. The SLATE implementation serves as the baseline in all our experiments.

## 5.2 Training slot attention with implicit differentiation

The first test to conduct is to see whether slot attention can be trained with implicit differentiation at all. *In summary:* **Test:** *Swap the forward pass with a different black-box solver, and train with implicit gradients computed via various gradient estimation methods for Eq. 1.* **Hypothesis:** *There exists a (forward solver, implicit gradient estimator) that optimizes cross entropy no worse than backpropagating through unrolled inference.* **Result:** *Yes, such a pair exists, and in fact it enables significantly better optimization.*

To conduct this test, we compare vanilla slot attention with four variants trained with implicit differentiation, shown in Table 2, labeled *IB*, *BB*, *IN*, and *BN*. B stands for the Broyden solver, used by Bai et al. [7]. We test configurations where the Broyden solver was both used to find

Table 2: Different instantiations of implicit differentiation

| abbrv. | forward computation | gradient estimation |
|--------|--------------------|--------------------|
| *IB* | iteration | Broyden |
| *BB* | Broyden | Broyden |
| *IN* | iteration | Neumann approximation |
| *BN* | Broyden | Neumann approximation |

the fixed point of slot attention and used to estimate the implicit gradient with Eq. 2. *N* stands for the first-order Neumann approximation for computing the explicit gradient (Eq. 3). Our hypothesis would be tentatively refuted if the cross-entropy learning curves of none of these variants converges as efficiently as the baseline. We train these methods to model the CLEVR-Mirrors dataset with an image resolution of 64x64. The slot attention baseline unrolls the slot attention cell for seven iterations and we also limit the maximum number of Broyden iterations to seven.

**Results and analysis** Figure 3 shows that not only is it possible to train slot attention with implicit differentiation, but three out of four of the implicit differentiation configurations from Table 2

---

[2]https://github.com/singhgautam/slate
[3]https://github.com/locuslab/deq

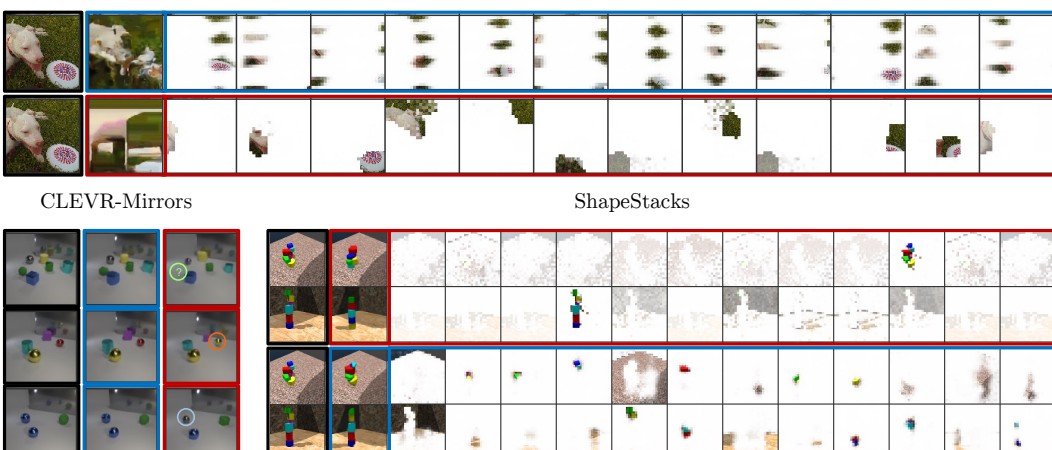

Figure 4: **Qualitative results.** Across three datasets, optimizing SLATE with implicit differentiation leads to improved image reconstructions through the slot bottleneck. Black borders indicate the ground truth image, blue border indicate our method, and red borders indicate vanilla SLATE. The rest of the panels visualize attention masks. In the CLEVR-Mirrors dataset, whereas vanilla SLATE misses objects, changes their size, or changes their color (indicated by the circles), implicit SLATE reconstructs the ground truth more faithfully.

optimize significantly better than vanilla slot attention. Whereas vanilla slot attention plateaus at a cross entropy of around 130, the *BB*, *IN*, and *BN* variants all achieve a 10x lower cross entropy loss. Moreover, *BB*, *IN*, and *BN* all follow the same learning curve, suggesting that the optimization improvement is largely due to whether we use implicit differentiation or not, rather than the choice of how we implement the implicit differentiation. The *IB* variant is less stable than the others, and we hypothesize that this can be explained by the tendency for the fixed point of Eq. 2 to become increasingly hard to estimate, an issue discussed in Bai et al. [8].

### 5.3 Does this improvement generalize across datasets?

We now test whether this improved optimization holds across different datasets. If the improvement was simply due to implicit differentiation being serendipitously useful for 64x64 images CLEVR-Mirror, then we should expect implicit differentiation to not help for other datasets. We focus our attention on *IN* variant to conduct this test because it is the minimal modification to the baseline slot attention for gaining the benefits of implicit differentiation, as it requires adding only one line of code on top of the baseline slot attention implementation and does not require implementing any blackbox solvers. We henceforth refer to the *IN* variant as **implicit slot attention** (and correspondingly implicit SLATE), as described in §4.3. *In summary: Test: Apply implicit slot attention to CLEVR-Mirror, ShapeStacks, and COCO with 96x96 image resolution. Hypothesis: Implicit slot attention significantly improves optimization across all three datasets. Result: Yes it does.*

**Results and quantitative analysis** Using the two primary metrics used in Singh et al. [57], images generated by implicit SLATE achieve both lower pixel-wise mean-squared error and FID score [29]. The FID score was computed with the PyTorch-Ignite [18] library using the inception network from the PyTorch port of the FID official implementation. All methods were trained for 250k gradient steps. Table 3 compares the FID and MSE scores of the images that result from compressing the SLATE encoder's set of discrete tokens through the slot attention bottleneck, using Image-GPT to autore-

Table 3: Quantitative metrics for image reconstruction through the slot bottleneck.

| Data | Implicit | Vanilla |
|---|---|---|
| CLEVR (FID) | **22.19** | 25.89 |
| CLEVR (MSE) | **10.66** | 67.04 |
| COCO (FID) | **127.79** | 147.48 |
| COCO (MSE) | **1659.15** | 1821.75 |
| ShapeStacks (FID) | **34.2** | 34.76 |
| ShapeStacks (MSE) | **108.67** | 312.14 |

gressively re-generate these image tokens one by one, and using the discrete VAE decoder to render the generated image tokens. Implicit differentiation significantly improves the quantitative image

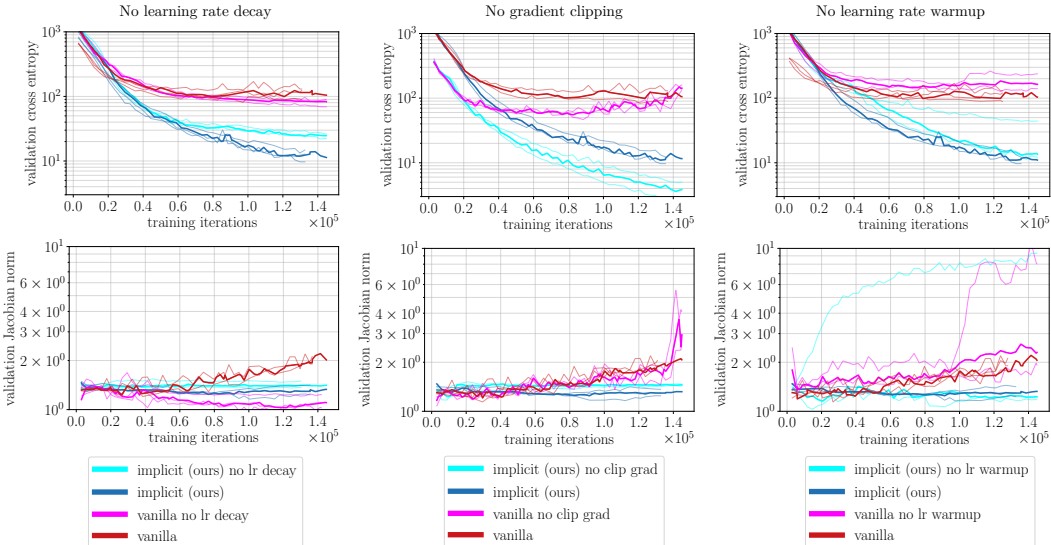

Figure 5: **Does implicit differentiation remove the need for various optimization tricks?** We ablate three heuristically-motivated optimization tricks from both vanilla SLATE and our method, and show that for two out of the three, removing the optimization trick quantitatively hurts the vanilla model but not the implicit model. Whereas removing gradient clipping and learning rate warmup causes vanilla SLATE's training to become unstable, as indicated by the growth of the Jacobian norm of the slot attention cell, our method trains significantly more stably and can take advantage of the larger gradient steps.

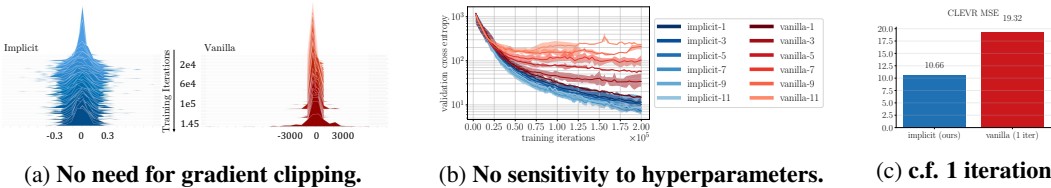

(a) **No need for gradient clipping.**     (b) **No sensitivity to hyperparameters.**     (c) **c.f. 1 iteration**

Figure 6: (a) Without gradient clipping, our implicit differentiation technique keeps gradients small while backpropagating through the unrolled iterations causes gradients to explode. (b) Training with implicit differentiation also is not sensitive to the number of iterations with which to iterate the slot attention cell. (c) Using one iteration for vanilla slot attention trains as stably, but reconstructs more poorly, than implicit slate.

reconstruction metrics of SLATE across the test sets of CLEVR-Mirrors, Shapestacks, and COCO. In the case of MSE for CLEVR, this is almost a 7x improvement.

**Qualitative analysis**  The higher quantitiatve metrics also translate into better quality reconstructions on the test set, as shown in Figure 4. For CLEVR-Mirrors, vanilla SLATE sometimes drops or changes the appearance of objects, even in simple scenes with three objects. In contrast, the generations produced from implicit SLATE match the ground truth very closely. For Shapestacks, implicit SLATE consistently segments the scene into constituent objects. This is sometimes the case with vanilla SLATE on the training and validation set as well, but we observed for both of the seeds we ran for the final evaluation that vanilla SLATE produced degenerated attention maps where one slot captures the entire foreground, and the background is divided among the other slots. The visual complexity of the COCO dataset is much higher than either CLEVR-Mirrors and Shapestacks, and the reconstructions on the COCO dataset are quite poor, for both SLATE's discrete VAE and consequently for the reconstruction through the slot bottleneck. This highlights the gap that still exists between using the state-of-the-art in object-centric learning out-of-the-box and what the community may want these methods to do. The attention masks for both vanilla and implicit SLATE furthermore do not appear to correspond consistently to coherent objects in COCO but rather patches on the image that do not immediately seem to match with our human intuition of what constitutes a visual entity.

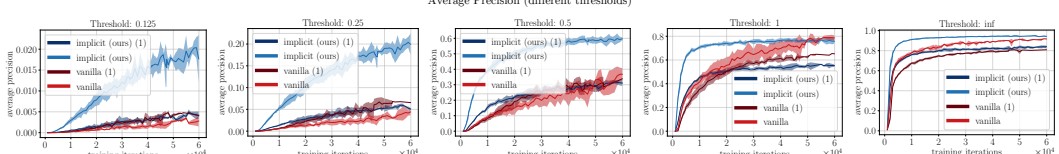

Figure 7: We outperform vanilla slot attention on object property prediction (see Locatello et al. [39]).

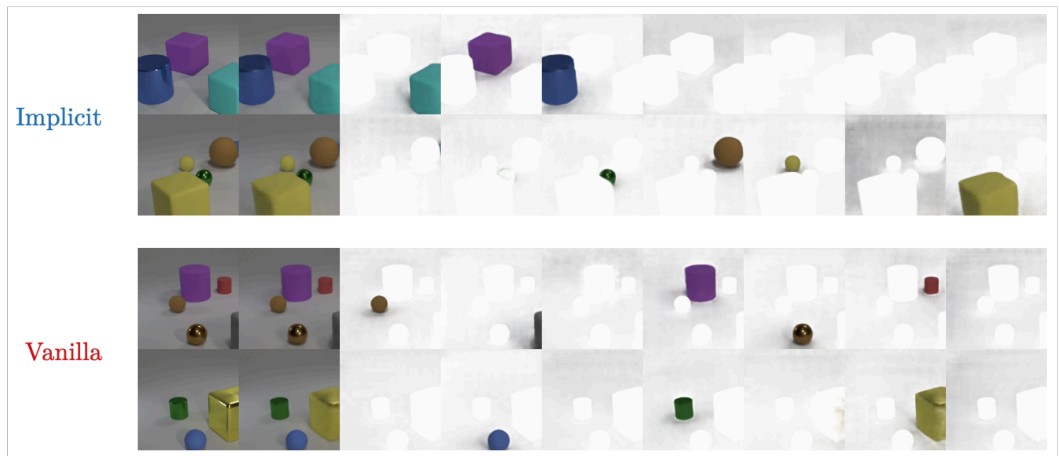

Figure 8: Implicit differentiation preserves the quality of predicted segmentations from Locatello et al. [39].

## 5.4 Can we simplify the need for optimization tricks?

To further understand the benefits of implicit differentiation, we then ask whether it stabilizes the training of slot attention without the need for optimization tricks like learning rate decay, gradient clipping, and learning warmup. *In summary:* **Test:** *Remove learning rate decay, gradient clipping, and learning warmup each from vanilla SLATE and our method.* **Hypothesis:** *Our method should not require these tricks to optimize stably.* **Result:** *No, it generally does not require these tricks.*

Fig. 5 shows that the ease and stability of training correlates with the spectral norm of the Jacobian of the slot attention cell. Decaying the learning rate regularizes the Jacobian norm from exploding, but it also hurts optimization performance for both our method and vanilla SLATE, as expected. When we remove gradient clipping the Jacobian norm of vanilla SLATE explodes, as do its gradients (Fig. 6a), whereas both stay stable for our method. This translates into a qualitative drop in performance for vanilla SLATE (Fig. 13) but not for implicit SLATE (Fig. 12). Lastly, removing learning rate warmup also consistently makes vanilla SLATE's training unstable, whereas it only affects the stability of our method for one out of three seeds. Finally, Fig. 6b shows that implicit slot attention is not sensitive to the number of iterations with which to iterate the slot attention cell, whereas vanilla slot attention is, with more iterations being harder to train.

## 5.5 Does this mean that iterating slot attention for one iteration is enough?

One might be tempted to interpret Fig. 6b to suggest that fewer iterations of vanilla slot attention is sufficient to improve performance. This is not necessarily the case: although vanilla slot attention with one iteration trains more stably than its seven iteration counterpart, it still achieves 2x worse reconstruction MSE than implicit slot attention with seven iterations on CLEVR (Fig. 6c).

Futhermore, Fig. 7 shows that using only a single iteration is not enough to improve optimization of vanilla slot attention for the object property prediction task used by Locatello et al. [39]. We directly modified the released code from Locatello et al. [39], which used three iterations, to use implicit differentiation. Implicit slot attention can instead scale to as many forward iterations as needed.

### 5.6 Does implicit slot attention still produce intuitive masks with a different architecture?

We sought to check whether implicit differentiation still preserves the quality of the segmentation masks produced by the original slot attention architecture by Locatello et al. [39], which uses a spatial broadcast decoder [63] rather than a transformer decoder as SLATE does. It indeed does (Fig. 8), suggesting that our findings are not specific to SLATE but apply to slot attention more broadly.

## 6 Discussion

The connection we made in this paper between slot attention and deep equilibrium models also highlights various other properties about iterative refinement procedures for inferring latent sets that suggest connections to other areas of research that are worth theoretically developing in the future. First is the connection to the literature on fast weights [6, 33, 54]: interpreting slots as parameters that are modified during the inner optimization during execution time may give us novel formulation for how to represent and update fast weight memories. Second is the connection to the literature on meta-learning [5, 17, 53, 60]: interpreting slots as solutions to an inner optimization problem during execution time may give us a novel perspective on perception as itself a learning process. Third is the connection to causality [46, 47]: interpreting the independently generated and symmetrically processed slots as parameterizing independent causal mechanisms [28] may give us a novel approach for learning to represent causal models within a neural scaffolding. Fourth is the connection to dynamical systems [44]: interpreting slots as a set of attractor basins may offer a novel theory of how the error-correcting properties of discrete representations emerge from continuous ones. These different fields have their own conceptual and implementation tools that could potentially improve our understanding of how to build better iterative refinement algorithms and inform how objects could potentially be represented in the mind.

**Conclusion**   We have proposed implicit differentiation for training iterative refinement procedures for inferring representations of latent sets. Our results show clear signal that implicit differentiation can offer a significant optimization improvement over backpropagating through the unrolled iteration of slot attention, and potentially any other iterative refinement algorithm, with lower space and time complexity and only one additional line of code. Because it is so simple to apply implicit differentiation to any fixed point algorithm, we hope our work inspires future work to leverage tools developed for implicit differentiation for improving learning representations of latent sets and iterative refinement methods more broadly.

## Acknowledgements

This work was supported ARL, W911NF2110097, and ONR grant number N00014-18-1-2873. Part of this work was completed while MC was an intern at Meta AI. Computing support came from Google Cloud Platform and Meta. We would like to thank Shaojie Bai for help on implicit differentiation, Gautam Singh for help on SLATE, Alban Desmaison for help on PyTorch. We would also like to thank Yan Zhang, David Zhang, Thomas Kipf, and Klaus Greff for insightful discussions and Jianwen Xie for pointing out previously missing references.

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
