# OpenReview forum: "Object Representations as Fixed Points: Training Iterative Refinement Algorithms with Implicit Differentiation"
_NeurIPS.cc/2022/Conference — NeurIPS 2022 Accept_

### Official Review · Reviewer_KLwZ · 2022-07-06

**Rating:** 6
**Confidence:** 3
**Soundness:** 3 good
**Presentation:** 3 good
**Contribution:** 2 fair

**Summary:**

The paper proposes to treat object-centric models with iterative refinement procedures as fixed point operations and optimize them using implicit differentiation. In their experiments, the authors show that the training of SLATE becomes more stable when truncating the gradients after the last refinement step instead of propagating it through the unrolled refinement procedure and show performance increases on three datasets (CLEVR, ShapeStack, COCO).


**Questions:**

Can this methods we applied to other object-centric models and if so, do the claims hold there?

**Strengths And Weaknesses:**

The paper is clearly structured, and the experiments nicely build on each other to support the claims of the paper.  Object-centric models are known to be brittle in training and with regard to hyperparameters. The authors show this problem and propose a solution. They convincingly show that slot attention can be trained using implicit differentiation while retaining/ improving its performance on synthetic datasets. As the authors rightfully state, object-centric model currently do not scale to natural datasets and are therefore of limited use at the moment. Unfortunately, the changed optimization procedure proposed here does not help with scaling to more natural data but fails to learn to identify objects in COCO.
Furthermore, I don’t see evidence that the optimization procedure helps with getting rid of optimization tricks. While implicit SLATE does show a more well-behaved Jacobian norm with and without learning rate decay, gradient clipping and learning rate warmup, it does not show a qualitatively different behavior when removing these tricks compared to vanilla SLATE. The results of the experiments rather seem to suggest that these tricks did not have a significant effect in the first place and could be in general removed (except maybe lr warmup).

Strength:
- Clear motivation of the work
- Well written and clearly structured
- Experiments on several datasets to support their claims including a natural image dataset


Weaknesses:
- The idea of using implicit differentiation to optimize neural networks is not new and the technical contribution limited.
- The paper would be stronger if the authors would show that their claims hold for more than one model.
- The model is still of very limited use since it only works on synthetic datasets.
- Experiments do not support claims regarding optimization tricks.
- Segmentation performance should be quantified in order to make a claim about it (Sec. 5.6)

---

> ### Author Response · Authors · 2022-08-02
> **Response to Reviewer KLwZ**
>
> We appreciate Reviewer KLwZ generally positive assessment of our work and helpful feedback, which greatly improved the paper. **To address reviewer KLwZ’s questions and concerns, we have updated the manuscript with results from one additional experiment: a qualitative comparison of the effect of removing gradient clipping.** Below we address the reviewer’s concerns and questions in detail:
> - **“Does not scale to COCO, only works on synthetic datasets”**: Our results on COCO show that improved and more stable optimization of SLATE does not enable better performance on more complex visual scenes, thus ruling out that difficulty in optimization is the primary cause of the difficulty in modeling complex visual scenes. This suggests the field may need new algorithms and new training settings to properly model complex scenes. We suggested a potential direction in Appendix A, which is to train object-centric models in embodied interactive settings.
> - **“No qualitative different behavior when removing optimization tricks”**: *We include an additional experiment showing that this concern is false.* In the updated submission, we show a visualization of the reconstruction quality and attention masks of implicit SLATE (FIgure 12) and vanilla SLATE (Figure 13), both trained without gradient clipping. We can see a clear difference between the quality of the reconstructions and masks. Implicit SLATE’s masks capture objects more coherently. Vanilla SLATE’s masks are much noisier, and become degenerate in the later stages of training as its Jacobian norm explodes.
> - **“Implicit differentiation for neural networks is not novel.”** Our methodological novelty is the insight that the iterative procedure used for symmetry-breaking in object-centric models reaches a fixed point, allowing various techniques for implicit differentiation to be used. Methods for implicit differentiation, including [16], are broadly applicable, and novel applications like ours of such techniques in other settings are prominent papers at high-profile venues in their own right (e.g. [24: Huang et al., NeurIPS 2021).
> - **“The paper would be stronger if the authors would show that their claims hold for more than one model.”** We in fact show that our technique works on multiple models (SLATE and original slot attention) for multiple tasks (image reconstruction, and object property prediction).
>
> Questions:
> - **Can this method be applied to other object-centric models and if so, do the claims hold there?** In informal experiments on SAVI [26] for videos, we found that our technique also improved the stability of training significantly. However, the scope of our paper focuses on object-centric models for static images, and video results would be much better suited for another paper.

---

> > ### Comment · Reviewer_KLwZ · 2022-08-05
> > **Thanks**
> >
> > I thank the authors for their response.
> >
> > **Optimization tricks:**
> >
> > The figure caption of Fig.5 reads “Implicit differentiation removes the need for many optimization tricks.”, which I think is misleading.  If that was the case, the vanilla model would need to perform much worse without the optimization tricks (compared to the vanilla model with the trick; which is not the case for lr decay and gradient clipping). For me the conclusion from the figure is that these tricks (lr decay and gradient clipping) weren’t really helpful in the vanilla model in the first place and one can remove them independently of using implicit differentiation.
> >
> > That the implicit model performs better than the vanilla model in absolute numbers is clear and is not what my comment was aiming at.
> >
> >
> > **Multiple models:**
> > The claim in the abstract/ introduction that many object-centric models have an iterative refinement process that would be stabilized by implicit differentiation reads rather broad. It would be more convincing if it was shown on different kinds of models, not only those that use SlotAttention (as SLATE and SAVI do).

---

> > > ### Author Response · Authors · 2022-08-05
> > > **Response**
> > >
> > > **Optimization tricks**: Our understanding of the reviewer's claim is that the vanilla model does not perform worse if we remove the optimization tricks, such as gradient clipping. Let us consider the case of gradient clipping. We showed in our revision (Figure 12 and Figure 13 of the appendix) that implicit differentiation *does* remove the need for gradient clipping. Figure 12 shows the reconstructions and attention masks for the implicit model *without* gradient clipping, and Figure 13 shows the reconstructions and attention masks for the vanilla model *without* gradient clipping. Figure 10 shows the reconstructions and attention masks for the implicit and vanilla models *with* gradient clipping.
> > > - It is clear from comparing Figure 13 and Figure 10 that removing gradient clipping creates a significant decrease in modeling quality for the vanilla model.
> > > - It is clear from comparing Figure 12 and Figure 10 that removing gradient clipping *does not* affect the modeling quality for the implicit model.
> > > - It is clear from comparing Figure 12 and Figure 13 that, when gradient clipping is removed, the implicit model produces significantly better reconstructions and attention masks than the vanilla model.
> > >
> > > If the reviewer is claiming that gradient clipping does not help the vanilla model, this experiment shows that this claim does not hold. If we have misunderstood the reviewer's concern, it would be helpful if the reviewer could clarify again.
> > >
> > > **Multiple models**. Thank you the comment. We can scope the claim more in the introduction. However, we note that the state-of-the-art models are all based on only SlotAttention, in the sense that SlotAttention has replaced Neural Expectation Maximization [18] and IODINE [19] as the method of choice. It just so happens that SlotAttention is only one state-of-the-art model at the moment, which is why we focused only on SlotAttention.

---

> > > > ### Comment · Reviewer_KLwZ · 2022-08-06
> > > > **Optimization tricks**
> > > >
> > > > My comment is based on Fig. 5 in the main paper, in which removing lr decay and gradient clipping does not decrease the performance of vanilla slate, but improves it. So there is a discrepancy between the figure caption and what is shown in the respective plots.
> > > > I'm rather surprised that you state that this is false. Could you clarify for me, if I'm misunderstanding Fig. 5 and it does not shown an improvement of vanilla slate in your chosen quantitative metrics when removing lr decay and gradient clipping compared to the vanilla model using the tricks?
> > > >
> > > > If there is different evidence to support the claim, i.e qualitative evidence, that is great and should be shown and then needs some explanation, why this is not reflected in the quantitative evaluation. If you don't feel like the chosen metrics measure what you want to show, maybe a different metric here would help.
> > > > I'm merely stating that what your results show and what you claim specifically in Fig. 5 is not well aligned, not that what your claiming is necessarily wrong.

---

> > > > > ### Author Response · Authors · 2022-08-08
> > > > > **Thank you for the clarification**
> > > > >
> > > > > Thank you for clarifying, and your comments have helped us tighten the claims of our paper.
> > > > >
> > > > > **Learning rate decay:** The reviewer is correct that removing learning rate decay does not decrease the performance of the vanilla model. We have clarified in the caption of Figure 5 that removing learning rate decay does not make a difference.
> > > > >
> > > > > **Gradient clipping:** our claim is that removing gradient clipping hurts the vanilla model, but does not hurt the implicit model. Figure 5 shows that the validation cross entropy curve for "vanilla no clip grad" (light pink) moves upward and crosses the validation cross entropy curve for "vanilla" (with gradient clipping). Simultaneously, we see that the Jacobian norm of "vanilla no clip grad" explodes. On the other hand, the light blue curve (implicit no clip grad) and the dark blue curve (implicit) do not cross. It is expected that the model without gradient clipping will train faster initially (because it gets larger gradients); the question is whether optimization without gradient clipping is stable enough for the model without gradient clipping to continue outperforming the model with gradient clipping. The model without gradient clipping *does not* continue outperforming the model with gradient clipping for the vanilla model, while the model without gradient clipping *does* continue outperforming the model with gradient clipping for the implicit model. Therefore, our quantitative results for gradient clipping are consistent with our qualitative results.
> > > > >
> > > > > **In summary**, we tested the removal of learning rate decay, gradient clipping, and learning rate warmup. For two out of the three (gradient clipping and learning rate warmup), removing these optimization tricks quantitatively hurts the vanilla model but does not quantitatively hurt the implicit model. We also show that removing gradient clipping makes a qualitative difference in performance for the vanilla model, but not the implicit model.

---

### Official Review · Reviewer_pdiR · 2022-07-09

**Rating:** 7
**Confidence:** 4
**Soundness:** 4 excellent
**Presentation:** 4 excellent
**Contribution:** 3 good

**Summary:**

Summary:
* The paper combines existing techniques from two groups of works. First, unsupervised object-centric learning with sets representation. Second, implicit differentiation techniques.

**Questions:**

* How do you expect the proposed approach to perform with more complex visual scenes / videos?
* Do you have any intuition of how changing Neumann series expansion affect he model? Will higher order approxmiation benefits the model?

**Limitations:**

I don't believe there is a limitation section explicitly in the main paper. But there are some in the supplementary.


**Strengths And Weaknesses:**


Strength:
* The paper is very nicely written overall. The logic flow take us through the author's thinking and reasoning clearly. Relevant sources are all cited at the correct places, making understanding the paper an enjoyable journey. The experiments and results are clearly presented with some helpful color-coding to aid the process. I really enjoy reading the paper.
* Though similar idea has been empirically explored in prior work [44], this paper does a much more fine-grained investigation, analysis and evaluation.
* Impressive empirical results are achieved by the techniques. The authors do a great job evaluating the performance quantitatively and qualitatively.


Weakness:
* While I do think the paper shows very interesting results, I don't think it's reasonable to motivate them with "Despite their conceptual elegance, it has been difficult to scale iterative refinement methods **beyond modeling simple static scenes or short video sequences** because differentiating through the unrolled forward iteration makes training unstable." The paper do not showcase beyond modeling simple static scenes or short video sequences (which is fine), but promising it in the intro makes it an empty promise.
* The techniques in implicit differentiation are by themselves not novel. The authors are applying existing techniques to a recurrent problem that has yet to see these techniques being applied.
* I personally think that mentioning that "in one instance improves reconstruction mean-squared error by almost 7x" feels like an oversale.
* If I'm not mistaken "BB" variant is basically applying Bai et al.'s DEQ [6] to the current setting. And from what I can tell from the training curve Fig 3b, they perform similarly. I think it might be interesting to show how previous technique performs quantitatively.

---

> ### Author Response · Authors · 2022-08-02
> **Response to Reviewr pdiR**
>
> We appreciate Reviewer pdiR’s generally positive assessment of our work and helpful feedback, which greatly improved the paper. **To address reviewer pdiR’s questions and concerns, we have updated the manuscript with results from two additional experiments: (1) quantitative results for the “BB” (Broyden solver) variant, and (2) a comparison of the various approximations to the Neumann series expansion.** Below we address the reviewer’s concerns and questions in detail:
> - **“Motivation of current methods’ ability to only model simple static scenes or short video sequences.”** Reviewer pdIR suggested that we adjust this motivation to better reflect our contribution as improving ease of optimization rather than unlocking new capability. We agree and have revised the paper to reflect this suggestion.
> - **“The techniques in implicit differentiation are by themselves not novel.”** We indeed do not propose a new technique for implicit differentiation. Instead, our methodological novelty is the insight that the iterative procedure used for symmetry-breaking in object-centric models reaches a fixed point, allowing various techniques for implicit differentiation to be used. Methods for implicit differentiation, including [16], are broadly applicable, and novel applications like ours of such techniques in other settings are prominent papers at high-profile venues in their own right (e.g. [24: Huang et al., NeurIPS 2021).
> - **BB variant**: *We have included quantitative results of the BB variant to Table 3.* We still observe overall that the first order Neumann approximation is generally the better method to use. Additionally, the Broyden solver is much more computationally expensive, running at 66% the speed of the IN variant.
>
> Questions:
> - **How do you expect the proposed approach to perform with more complex visual scenes and videos?** Our results on COCO show that improved and more stable optimization of SLATE does not enable better performance on more complex visual scenes, thus ruling out that difficulty in optimization is the primary cause of the difficulty in modeling complex visual scenes. This suggests the field may need new algorithms and new training settings to properly model complex scenes. We suggested a potential direction in Appendix A, which is to train object-centric models in embodied interactive settings. In informal experiments on SAVI [26], we found that our technique also improved the stability of training significantly.
> - **How does changing the Neumann series expansion after the model?** *We have included an additional comparison of the various approximations to the Neumann series expansion in Figure 11 of the updated submission.* We observe that the 1st order approximation still largely performs the best, likely because adding more terms to the series expansion requires backpropagating through more iterations of slot attention, which was the problem we had sought to avoid in the first place. However, most approximations still perform better than the vanilla model with the same number of forward iterations.

---

### Official Review · Reviewer_qYDF · 2022-07-11

**Rating:** 8
**Confidence:** 4
**Soundness:** 4 excellent
**Presentation:** 4 excellent
**Contribution:** 4 excellent

**Summary:**

This paper notes that the a number of state-of-the-art object-centric learning models rely on an iterative refinement step that can be understood as a fixed point algorithm. It then uses implicit differentiation to avoid a costly and unstable unrollment to increase the performance of SLATE, a state-of-the-art object-centric learning method, on a range of metrics and datasets with minimal changes to the code.

**Questions:**

- On Coco 2017 (Fig. 4), the solutions with iterative refinement seem to settle on a very different minimum: while standard SLATE finds pixel blobs in each slot (but doesn't reach good reconstruction), implicit SLATE seems to find stripes of small blobs. Do you have an explanation for this behaviour?
- In Figure 5, what dataset are you using here?

**Limitations:**

I don't see potential negative impacts.

**Strengths And Weaknesses:**

- Originality: At its core, the technical innovation itself is limited to transferring existing methods from implicit differentiation to object-centric learning.
- Quality: The range of experiments is excellent, with carefully chosen modifications from the original code of previous methods to ensure reliable and fair comparisons.
- Clarity: The manuscript was one of the most enjoyable reads in a while, and I complement the authors for their careful, structured and clear presentation of their methods, the background and the experiments.
- Significance: While technically simple, the insight yields how performance gains that make it likely that this method will become a standard in the further development of the field.

---

> ### Author Response · Authors · 2022-07-27
> **Response to Reviewer qYDF**
>
> We appreciate Reviewer qYDF’s generally positive assessment of the significance, quality, and clarity of our work. The originality of our work lies in our key insight that the iterative procedure used for symmetry-breaking in models with latent sets reaches a fixed point, thus allowing implicit differentiation to be used. The scope of our method more broadly applies to models with latent sets, as discussed in Section 4; it just so happens that object-centric learning has historically been the primary application of models with latent sets. We have scoped our contribution to iterative refinement of latent sets, but it is a contribution that advances the optimization of the state-of-the-art. Object-centric learning is also beginning to be applied in model-based reinforcement learning [40] and video modeling [26], and we expect that in future work our method could potentially improve optimization in those areas as well.
>
> Questions:
> - **Different qualitative behavior on COCO 2017**. It’s not well understood why there is a qualitative difference in COCO, mostly because it is not well understood how to get object-centric methods to decompose realistic images yet in general (an analysis on the qualitative effects of implicit differentiation may be more meaningful once the baseline method is better at modeling more realistic images). For example, see the Appendix for a qualitative analysis on the differences between implicit SLATE and vanilla SLATE on CLEVR. In the CLEVR setting, we can perhaps suggest a more concrete explanation for the qualitative differences, which relies on the interpretation of slot attention as a differentiable approximation to the EM algorithm (Section 4.1): since the slot attention cell is trained to compute a single EM update, stopping gradients from flowing through the entire EM update process means that the slot attention cell in implicit slot attention is less regularized, which may result in less convex-blob-like attention masks.
> - **Figure 5 dataset**: This is CLEVR-Mirrors dataset from SIngh et al.

---

> > ### Comment · Reviewer_qYDF · 2022-08-05
> > **Thanks**
> >
> > I thank the authors for their response. After reading the other reviews and how their concerns are addressed, I am going to stick with my (positive) evaluation.

---

### Official Review · Reviewer_STEd · 2022-07-12

**Rating:** 5
**Confidence:** 3
**Soundness:** 3 good
**Presentation:** 3 good
**Contribution:** 2 fair

**Summary:**


The paper deal with the problem of objections (or object centric) learning by using differentiable iterative refinement to improve the stability and tractability during training. By that, the method seems to improve the stability of a vanilla slot attention architecture such as SLATE. The paper evaluates the proposed extension on the CLEVR-Mirror, Shapestacks, and COCO-2017 datasets.

**Questions:**


I'm not an expert in the field, so I would be one of the readers who would highly profit from a broader introduction of the topic and a bit more context with respect to related works in the application area and comparison to respective benchmarks.

I get and like the idea of implicit differentiation, but I#m wondering if objectness in slot attention is really the only use case here.

**Limitations:**

The method seems to be very generic. Therefore, I don't see any direct limitations and potential negative societal impact.

**Strengths And Weaknesses:**

Strengths:

- The paper addresses a not so well explored topic, but nevertheless important and interesting topic that would justify more research.

- The idea of extending the vanilla SLATE by a differentiable component during training seems interesting on a high level, but I'm not an expert, so it would be ideal to consider feedback from someone with hands-on experience in this domain.

- The paper provides an extensive evaluation of the different aspects of the proposed method.

Weaknesses:

- No coverage of related work in object discovery or unsupervised object detection:
There seems to be a decent body of work on this topic, that's not covered in the paper (e.g. for some older work: Pixel Objectness: Learning to Segment Generic Objects Automatically in Images and Videos, Bo Xiong et al. TPAMI 2018). There is also no comparison to any of those methods.

- No comparison to related work:
It would be good to see and understand the general state-of-the-art in the field and how the proposed method compares to this on some benchmarks.

- Limited scope?
Only framing this paper as a valid SLATE extension would be a bit narrow and might serve only a very small niche. If Slot attention architecture has a wider application that this idea might benefit from, it would be good to mention it and show the impact on respective experiments.

---

> ### Author Response · Authors · 2022-07-27
> **Response to Reviewer STEd**
>
> We thank Reviewer STEd for their thoughtful review. Below we address each of their concerns in turn.
> - **Coverage of related work**: Reviewer STEd is concerned that there is “no coverage of related work in object discovery or unsupervised object detection”. However, we in fact do conduct an extensive coverage [18–20, 26, 28, 37, 39, 40, 44] in both the introduction and related works sections.These cited works serve to provide a broad introduction to the research area that our paper contributes to (please see the survey [20] (https://arxiv.org/abs/2012.05208, which we have already cited in the related work). Reviewer STEd suggests Xiong et al. as a point of comparison, but Xiong et al. actually do not consider the same problem setting as us: Xiong et al. uses image annotations, whereas our method, and those methods we cited, are completely unsupervised.
> - **Comparison to state-of-the-art**: Reviewer STEd stated that we did not compare with the state-of-the-art. However, we in fact do compare to the state-of-the-art for unsupervised object discovery, SLATE, as we stated in the last paragraph of the introduction. We also compared with the predecessor to SLATE, which is the original slot attention architecture, in Figure 7 and Figure 8.
> - **Limited scope**: Reviewer STEd asked for impact on wider applications beyond extending vanilla SLATE. However, we do already show that our contribution extends beyond simply just image reconstruction, which was the objective of vanilla SLATE, to other applications such as set property prediction (Figure 7). As such, we have applied our contribution to multiple applications, each of which have been the focus of individual past publications. For example, we show our method for image reconstruction (c.f. [37], ICLR 2022) in Table 3, unsupervised object decomposition (c.f. [28]) in Figure 8, and set-property prediction (c.f. [43, 28]) in Figure 7.
>
> We now address Reviewer STEd’s questions:
>
> - **Context with respect to related works**: As stated above, we have discussed our contribution in the context of an extensive literature review [18–20, 26, 28, 37, 39, 40, 44] in both the introduction and related works sections. Please see the survey [20] for an overview of the research area.
> - **Other use cases?** As stated above, we have shown that our contribution is not simply limited to “objectness” and applies to at least three application areas:  image reconstruction (c.f. [37], ICLR 2022) in Table 3, unsupervised object decomposition (c.f. [28]) in Figure 8, and set-property prediction (c.f. [43, 28]) in Figure 7.

---

> > ### Author Response · Authors · 2022-08-04
> > **Have we resolved your concerns?**
> >
> > Hi Reviewer STEd,
> >
> > We just wanted to post a friendly reminder to ask if we have resolved your original concerns, and whether there are additional clarifications or explanations we can provide.

---

> > > ### Comment · Reviewer_STEd · 2022-08-08
> > > **Response for Paper11239**
> > >
> > > Dear authors,
> > >
> > > Thanks so much for the explanation.
> > >
> > > Coverage of related work - I agree that there is coverage from the ML side, but with “no coverage of related work in object discovery or unsupervised object detection” I was mainly referring to a parallel stream of ideas in the computer vision community, where the same idea is tried but with different settings in mind. The survey in [20] (On the Binding Problem in Artificial Neural Networks) does cover this a bit in 4.3, but is far away of going deeper into methods and benchmarks outside of ML methods. However, if there is no interest in this line of work, I would leave it to the authors how to place the paper in the community.
> > >
> > > Comparison to state-of-the-art - With comparison to state-of-the-art, I would refer to a direct comparison to the previously published numbers tasks, and benchmarks. E.g. The SLATE paper reports FID and MSE for 7 different datasets. The current evaluation uses only one of those and does not even include the numbers reported in the SLATE paper (CLEVR FID 37.42, MSE 37.42 for image reconstruction on CLEVR).
> > > There are also no comparison reported from any other approaches including the original slot attention paper, which tested Adjusted Rand Index (ARI) on CLEVR6, Multi-dSprites, and Tetrominoes, or any other method in the field, e.g. IODINE, MONet, GENESIS, GENESIS-V2 (GENESIS-V2: Inferring Unordered Object Representations without Iterative Refinement, Martin Engelcke et al., NeurIPS 2021) etc.  GIRAFFE would also be interesting, but I would understand is this is too far out (GIRAFFE: Representing Scenes as
> > > Compositional Generative Neural Feature Fields, Michael Niemeyer et al., CVPR 2021). More works are also listed in [20] under 4.3.3. Note that I'm not requesting to compare to all of them, but a decent selection of previous methods >=2 (so more than just SLATE, including a direct comparison to numbers from previous papers) usually helps to put a paper in a larger context and shows how the field evolved and allows for direct comparability of methods. Previous publications in the field have done this in various ways, so I assume that the topic is mature enough to ask for such a thing.
> > >
> > > Limited impact - For set-property prediction, AP on CLEVR10 seems to be already very high for vanilla slot attention[28] and numbers reported in [43] confirm this impression. It would be good to have a direct comparison here. I get it that training curves are an important part of the evaluation, but to really make a point here and allow others to build on top of that, concert metrics would allow for a direct comparison on this task.
> > > In general, this point is not about the number of downstream tasks (not saying that this can't help), but about the question who's research would profit from publishing this work. I see that this could be a stepping stone in an interesting direction, but what's currently still a bit too narrow for a top conference is the embedding of the proposed method within current state-of-the-art (including comparisons to the diverse ecosystem of other methods out there) as well as the anchor points in terms of reporting metrics that would allow other papers to compare to the proposed method directly. So overall, instead of growing this interesting field and allowing people to draw from it, it feels more like it is reduced.

---

> > > > ### Author Response · Authors · 2022-08-08
> > > > **Thank you for your response**
> > > >
> > > > Thank you for your response.
> > > >
> > > > **"The current evaluation uses only one of those and does not even include the numbers reported in the SLATE paper"**:
> > > > - We want to clarify that we compared to two (not one) of datasets in the original SLATE paper: CLEVR and ShapeStacks. At the time of writing, the SLATE paper (https://openreview.net/references/pdf?id=QIw5IbQlSi) only included 4 datasets: 3D shapes, CLEVR, ShapeStacks, and Bitmoji, and we chose to not to focus on 3D shapes because it was the most toy of the datasets, and we chose to not focus on Bitmoji because Bitmoji was restrictive in the full range of compositionality of objects (for each scene there can only be one pair of eyes, one nose, one mouth for example). We did not consider CelebA for the same reason. We did not evaluate on textured MNIST or ClevrTex because the code used for this dataset was not included in the official repo (see section A.3 of the final SLATE paper).
> > > > - We also directly used the code provided by the SLATE authors (https://github.com/singhgautam/slate) and reached out to the SLATE authors reporting a discrepancy in the quantitative performance we obtained from running the code downloaded from the SLATE repo and the performance reported in the SLATE paper. We did not report the numbers reported in the SLATE paper because that was not the numbers we got from running the official code from the SLATE repo, and so reported the numbers we obtained to be a consistent reference for the numbers for the implicit model.
> > > >
> > > > **"There are also no comparison reported from any other approaches"**
> > > > - Slot attention was shown to outperform IODINE, so we compared to Slot Attention as the state of the art.
> > > > - MONET, GENESIS, GENESIS-V2, GIRAFFE do not use iterative refinement, whereas the specific hypothesis our paper is testing is whether implicit differentiation improves the training of iterative refinement methods. Therefore, these models are not relevant to the question our paper is asking. Note that neither SLATE nor Slot Attention had compared to MONET, GENESIS, GENESIS-V2, GIRAFFE either.
> > > >
> > > > **"Direct comparison for set-property prediction"**
> > > > -  We directly downloaded the official code provided by the Slot Attention authors and directly reported numbers from (1) running their code verbatim and (2) running the implicit version. Therefore, this is a direct comparison because we directly used the official code. Our claim is about optimization stability, which is why the training curves, rather than the final evaluation is more relevant to the hypothesis we are testing.
> > > >
> > > > **Summary**
> > > > - Our implicit model advances the training of the state-of-the-art (SLATE and Slot Attention) object-centric methods.
> > > > - Our claim is about training stability and optimization performance, and our experiments were chosen to test this claim.
> > > > - Our comparisons were direct, because we directly used the official code provided by the authors of SLATE and Slot Attention and made only the modification we have proposed in Figure 2.

---

> > > > > ### Comment · Reviewer_STEd · 2022-08-09
> > > > > **Response for paper 11239**
> > > > >
> > > > > You did a good job defending the paper, so I’m raising my score to 5. Overall please consider the points raised in the future. This is not about devaluing the paper, but showing points for improvement. I understand that the current setting is not made for that, so probably read my comments again when the paper will be published.

---

> > > > > > ### Author Response · Authors · 2022-08-09
> > > > > > **Thank you for your feedback**
> > > > > >
> > > > > > We thank Reviewer STEd for their constructive feedback, which has raised many helpful points of investigation for future iterations of work for improving object-centric models with implicit differentiation.

---

### Meta-Review · Area_Chair_2Upv · 2022-08-22

**Recommendation:** Accept
**Confidence:** Less certain

**Metareview:**

The paper proposes to treat object-centric models with iterative refinement procedures as fixed point operations and optimize them using implicit differentiation.

Overall, the reviewers find that the contribution of the paper is somewhat novel, although similar ideas have been presented in prior work in different contexts (supervised settings). Only one reviewer was more negative before the rebuttal, eventually increasing their score after discussion with the authors.

I, therefore, recommend acceptance and encourage the authors to address the comments raised by the reviewers in the final version.

**Award:**

No

---

### Decision · Program_Chairs · 2022-09-14

Accept